# Deferoxamine B: A Natural, Excellent and Versatile Metal Chelator

**DOI:** 10.3390/molecules26113255

**Published:** 2021-05-28

**Authors:** Denise Bellotti, Maurizio Remelli

**Affiliations:** Department of Chemical, Pharmaceutical and Agricultural Sciences, University of Ferrara, 44100 Ferrara, Italy; blldns@unife.it

**Keywords:** deferoxamine B (DFOB), metal complexes, chelation therapy, solution equilibria

## Abstract

Deferoxamine B is an outstanding molecule which has been widely studied in the past decade for its ability to bind iron and many other metal ions. The versatility of this metal chelator makes it suitable for a number of medicinal and analytical applications, from the well-known iron chelation therapy to the most recent use in sensor devices. The three bidentate hydroxamic functional groups of deferoxamine B are the centerpiece of its metal binding ability, which allows the formation of stable complexes with many transition, lanthanoid and actinoid metal ions. In addition to the ferric ion, in fact, more than 20 different metal complexes of deferoxamine b have been characterized in terms of their chemical speciation in solution. In addition, the availability of a terminal amino group, most often not involved in complexation, opens the way to deferoxamine B modification and functionalization. This review aims to collect and summarize the available data concerning the complex-formation equilibria in solutions of deferoxamine B with different metal ions. A general overview of the progress of its applications over the past decade is also discussed, including the treatment of iron overload-associated diseases, its clinical use against cancer and neurodegenerative disorders and its role as a diagnostic tool.

## 1. Introduction

Deferoxamine B (DFOB, Scheme 1) is a natural siderophore, originally discovered in a soil bacterium, *Streptomyces pilosus*, but also produced by several species of both terrestrial and marine actinomycetes, often together with other siderophores [1]. In general, the biological function of a siderophore is to sequester, from the environment or from the host, Fe(III), which is available in aqueous solution, under aerobic conditions and neutral pH only at a very low concentration (10^−18^ mol·dm^−3^, approximately [2]). It is accepted that the biosynthesis of siderophores is activated when the intracellular iron concentration falls below the lower growth limit of the bacterium, estimated at 1 micromolar [3]. Once the complex between the siderophore and Fe(III) is formed, it returns to the generating cell through specific recognition pathways that include surface proteins capable of activating an import sequence. The special interest in DFOB derives from its important pharmacological role for patients with blood diseases that depend on transfusions (such as β-thalassemia major) and hemochromatosis. These patients must regularly assume DFOB (marketed under the trade name Desferal^®^), either alone or in combination with other drugs, to remove excess iron due to the hemolysis of blood cells. The human body does not have a mechanism for eliminating excess iron and the metal accumulates in the organs, especially in the heart and liver, with truly devastating toxic effects. The life expectancy of these patients has been enormously extended since the introduction of chelation therapy. The very high affinity of DFOB for Fe(III)—and for hard metal ions in general—derives from the fact that its linear structure contains three bidentate hydroxamic functional groups that wrap around the ferric ion to form a neutral and very stable octahedral complex (feroxamine B, FOB, Scheme 1). DFOB can remove iron only from storage proteins (ferritin and hemosiderin) and not from hemoglobin and transferrin, hence preventing undesirable clinical progressions.

The ability of DFOB to bind Fe(III) and many other metal cations, the large availability of data regarding its biological activity, its toxicity, its long-term effects on the human body (due to its prolonged use as a chelating drug), its acceptable solubility in water, the not excessive molecular weight and its relatively low cost have opened the way for the use of DFOB in various fields [4], such as: the treatment of diseases associated with the dyshomeostasis of metal ions; diagnostic imaging based on radioactive metals, the inhibition of processes that depend on metals, such as the so-called “nutritional immunity” through which the host organism seizes the microelements necessary for the development of invading microorganisms with the aim of preventing their growth, and the transfer of drugs to target cells by exploiting the natural access routes and recognition pathways of the FOB complex. The development of these drugs is made possible by the presence of the terminal amino group which is not involved in the metal coordination and allows DFOB to link to small molecules, antibiotics, antibodies or other compounds of biological interest, using simple chemistry to produce derivatives with new properties.

This review aims to summarize the available data concerning the complex-formation equilibria in solution of DFOB with metal ions and to make a general overview on the possible applications of this molecule in the different fields of medicinal chemistry.

## 2. Solution Equilibria Studies

### 2.1. DFOB Protonation

DFOB contains one protonable primary amino group and three hydroxamic groups which behave as very weak acids. Following the nomenclature normally used in the study of solution equilibria, the free ligand, in its completely deprotonated form, will be indicated as L^3−^. At neutral pH, DFOB is still fully protonated and is found in its H_4_L^+^ form; as the pH increases, the hydroxamic protons are released and then, lastly, the proton bound to the amine, which is the most basic group of the molecule. Table 1 shows the partial protonation constants of DFOB available in the literature and measured under different experimental conditions of temperature (*T*) and ionic strength (*I*); a representative distribution diagram is shown in Figure 1. It is not possible to highlight from the table relevant trends, as the experimental conditions vary. Curiously, protonation constants at 37 °C and an ionic strength of 0.15 mol·dm^−3^ (NaCl), certainly useful in the study of DFOB behavior under physiological conditions, have never been measured to the authors’ knowledge.

### 2.2. Fe(III)/DFOB Complexes

Although the hydroxamic groups are very weak acids and, in the absence of metal ions, are completely protonated in acidic or neutral solution, their affinity for the Fe(III) ion is so high that, in an equimolar solution, all of the metal ions are already “seized” by DFOB at pH 2 (Figure 2). For this reason, potentiometry is not suitable to measure the formation constants of the most protonated species. Exploiting the intense color of the Fe(III)/DFOB complexes, very reliable results have been obtained using a mixed spectrophotometric/potentiometric experimental method, which allowed the researchers to extend the investigation range below the reliability pH limits of the glass electrode [13]. The dominant species in the pH range 1–10 is the complex [Fe(III)HL]^+^, where the terminal amino group is protonated and not involved in complexation. The deprotonation of this group is characterized by a p*K*_a_ value of 10.7, very similar to that measured for the free ligand; this confirms that the amino group does not interact with the metal ion. The formation constants of the Fe(III)/DFOB complexes available in the literature are collected in Table 2.

The dissociation kinetics of the [Fe(III)HL]^+^ complex, as pH decreases from 1.52 to about 0, have been studied at 25 °C and at an ionic strength of 0.2 mol·dm^−3^ (HClO_4_/NaClO_4_) [14]. The separation of DFOB from Fe(III) involves a total of nine different species; the unwrapping process starts from the protonated amino terminal with the progressive detachment of the hydroxamic oxygens which are replaced by water molecules.

The hypothesis of amino group protonation at neutral pH is in agreement with the crystalline structure of FOB [15], in which the six oxygen atoms of the hydroxamate groups bind the iron in a distorted octahedral structure, forming two large rings chelates; the side chain containing the protonated terminal amine stretches away from the metal ion. Other crystallographic investigations on the interaction of FOB with proteins devoted to the transport of siderophores, such as FhuD of *Escherichia coli* [16] and FhuD2 of *Staphylococcus aureus* [17], confirmed that the terminal amino group can be exploited to derivatize DFOB without hindering its biological activity. The affinity of FOB for these proteins, measured in terms of the dissociation constant *K*_D_, is 0.90 and 0.05 μmol·dm^−3^, respectively [18,19]. FOB occupies the appropriate cavity of the transport protein, with the terminal amino group oriented outside the protein and therefore available to bind a chromophore, a fluorophore or other molecules of which FOB can act as a vector.

**Table 2 molecules-26-03255-t002:** Overall complex-formation constants (logβ) of DFOB with the Fe(III) ion under different experimental conditions. The constant in parentheses corresponds to the formation of the 1:1 complex of Fe(III) with DFOB protonated at its amino group and therefore refers to the equilibrium: Fe^3+^ + HL^2-^ = MHL^+^.

[MH_2_L]^2+^	[MHL]^+^	[ML]	*I* (mol·dm^−3^)	Background Electrolyte	*T* (°C)	Ref.
42.54	41.6	(30.60)	0.10	NaNO_3_	20.0	[5]
42.33	41.39	30.99	0.10	KCl	25.0	[7]
42.4	41.01	30.4	0.20	KCl	25.0	[20]
42.82	41.80	31.10	1.00	NaClO_4_	25.0	[13]

### 2.3. DFOB Complexes with Divalent Metal Ions

According to the Pearson acid-base concept, alkaline earth metals are considered hard acids; in fact, they can bind the oxygen donor atoms of DFOB and form variously protonated mononuclear complexes (see Table 3). The thermodynamic stability decreases going down the group, as the ionic radius increases; in the case of Be(II), the formation of a trinuclear hydroxyl complex has been hypothesized [21].

Complex-formation equilibria of DFOB with some divalent transition metal ions were also investigated. The results are summarized in Table 4; the data for the same cation but taken from various sources are in fair agreement with each other, considering the different experimental conditions. As regards the hard/soft character, the divalent metals of the first transition series are generally considered intermediate (or borderline) and they bind DFOB less strongly than Fe(III). All these cations form variously protonated mononuclear complexes with an affinity for the ligand following the Irving–Williams series. Protonated species correspond to complexes in which not all hydroxamic groups are bonded to the metal. The terminal amino group never interacts with the metal ion and releases its proton at basic pH, as occurs for the free ligand. Very stable complexes are formed by Cu(II), which is also the only metal in the series capable of forming a binuclear species. It is evident from these data that DFOB-based therapies can interfere with the metabolism of these essential metal ions [22].

Additionally, Cd(II) can bind DFOB, although with a lower affinity with respect to the metals of the first transition series [5].

Sn(II) and Pb(II) show good affinity for DFOB, especially the former, which has a harder character than Pb(II). Complexes of Sn(II) with DFOB are also more stable than those of Cu(II), and the formation of two binuclear species has been proposed [9].

### 2.4. DFOB Complexes with Trivalent Metal Ions

Other trivalent metal ions can form stable complexes with DFOB, as shown in Table 5. Only mononuclear and 1:1 species have always been detected. The most stable complexes are those formed by Co(III): noteworthily, the corresponding stability constants are more than five orders of magnitude higher than those of Fe(III). This result is confirmed by the very short M-O bond distances both measured by EXAFS and calculated by density functional theory (DFT) for the Co(III)/DFOB complex [23,24]. Very high affinity for DFOB is also shown by Mn(III) and Bi(III), but it is worth underlining that, in the case of the latter metal, the values of the constants are not fully reliable [9], due to the strong tendency of bismuth to hydrolyze even at very low pH. In the case of Al(III), it has been proposed [25] to exploit the stability of the complexes with DFOB for the treatment of aluminum intoxication, such as the consequences of dialysis with insufficiently purified water. However, the simple use of DFOB as a chelating drug for Al(III) will seriously interfere with iron metabolism. As for gallium, its complexes with DFOB have been studied for application in diagnostic imaging with the positron emission tomography (PET) technique, using the isotope ^68^Ga [26]. In the series of the 13th group, the stability of the complexes with Al(III), Ga(III) and In(III) decreases as the ionic radius increases, as expected. Table 5 also reports the only data available in the literature relating to DFOB complexes with lanthanides; they are rather weak, due to the high cation radius.

**Table 4 molecules-26-03255-t004:** Overall complex-formation constants (logβ) of DFOB with divalent transition metal ions under different experimental conditions. The constants in parentheses correspond to the formation of the 1:1 complex of M(II) with DFOB protonated at its amino group and therefore refer to the equilibrium: M^2+^ + HL^2−^ = MHL.

	[MH_3_L]^2+^	[MH_2_L]^+^	[MHL]	[ML]^−^	[M_2_L]^+^	[M_2_HL]^2+^	*I* (mol·dm^−3^)	Background Electrolyte	*T* (°C)	Ref.
Mn(II)	-	25.9	17.8	7.5	-	-	0.10	NaCl	25.0	[11]
Fe(II)	33.53	27.90	-	-	-	-	0.10	NaNO_3_	20.0	[5]
Co(II)	33.91	28.06	21.31	(10.31)	-	-	0.10	NaNO_3_	20.0	[5]
	34.00	27.80	20.2	9.4	-	-	0.10	NaCl	25.0	[24]
Ni(II)	34.09	28.40	21.90	(10.90)	-	-	0.10	NaNO_3_	20.0	[5]
	33.20	27.66	19.71	8.89	-	-	0.20	KCl	25.0	[10]
Cu(II)	37.33	34.20	25.12	(14.12)	-	33.42	0.10	NaNO_3_	20.0	[5]
	37.18	33.55	24.63	(13.73)	-	32.20	0.10	NaClO_4_	20.0	[9]
	37.07	33.38	24.41	(13.54)		32.12	0.10	NaClO_4_	25.0	[9]
	36.99	33.10	23.98	13.73	-	32.09	0.20	KCl	25.0	[10]
Zn(II)	34.18	28.57	22.07	(11.07)	-	-	0.10	NaNO_3_	20.0	[5]
	33.71	28.32	20.44	(9.55)	-	-	0.10	KNO3	25.0	[9]
	33.40	28.17	20.40	10.36	-	-	0.20	KCl	25.0	[10]
Pb(II)	35.34	29.69	20.89	(10.00)	-	27.18	0.10	KNO_3_	25.0	[9]
Cd(II)	33.05	26.28	18.88	(7.88)	-	-	0.10	NaNO_3_	20.0	[5]
Sn(II)	43.46	40.76	32.01	21.90	37.72	42.05	0.10	KCl	25.0	[9]

### 2.5. DFOB Complexes with Zr(IV)

The recent interest in the use of ^89^Zr as the positron emitting isotope in PET has prompted intense research into complexing agents that are able to both chelate the zirconium ion in a stable manner and bind a biological vector, such as an antibody or a nanoparticle. The ideal ligand for zirconium must be able to prevent metal hydrolysis, transchelation and demetalation in the environment before it is excreted. The Zr(IV) ion is characterized by a small ionic radius and a high charge and this makes it suitable for binding oxygen-donors like in siderophores [27,28]. The preferred coordination number of Zr(IV) is eight. There are no solid state structures of the Zr(IV) complexes with DFOB, but recent DFT calculations have verified the stability of a structure in which the three hydroxamate groups of DFOB are bonded to zirconium together with two water molecules in cis position of a distorted square antiprismatic structure [29,30].

**Table 5 molecules-26-03255-t005:** Overall complex-formation constants (logβ) of DFOB with trivalent transition metal ions under different experimental conditions. The constants in parentheses correspond to the formation of the 1:1 complex of M(III) with DFOB protonated at its amino group and therefore refer to the equilibrium: M^3+^ + HL^2−^ = MHL^+^.

	[MH_3_L]^3+^	[MH_2_L]^2+^	[MHL]^+^	[ML]	*I* (mol·dm^−3^)	Background Electrolyte	*T* (°C)	Ref.
Al(III)	-	35.11	33.93	24.50	0.10	KCl	25.0	[7]
	-	36.6	33.8	23.9	0.20	KCl	25.0	[31]
Ga(III)	-	-	38.96	28.65	0.10	KCl	25.0	[7]
	39.58	38.8	37.7	(27.56)	1.00	KCl	25.0	[6]
	-	-	36.92	27.56	0.20	KCl	25.0	[32]
In(III)	-	34.54	31.39	21.39	0.10	KCl	25.0	[7]
	-	36.40	32.48	22.18	0.20	KCl	25.0	[32]
Mn(III)	-	-	-	(28.6)	0.10	NaCl	25.0	[11]
Co(III)	-	-	-	(36.1)	0.10	NaCl	25.0	[24]
La(III)	34.33	28.31	21.89	(10.89)	0.10	NaClO_4_	25.0	[5]
Yb(III)	36.13	31.7	27.0	(16.0)	0.10	NaClO_4_	25.0	[5]
Bi(III)	-	≈40.0	≈36.2	(≈25.3)	0.10	NaClO_4_	25.0	[9]

The great tendency towards hydrolysis of zirconium in aqueous solutions is a significant obstacle to the thermodynamic study of these systems and only very recently the problem has been independently addressed by two research groups with results that are not entirely in agreement with each other, as shown in Table 6.

The high affinity of Zr(IV) for hydroxamate groups causes the metal to be fully complexed in the presence of an equimolar amount of DFOB already at a pH as low as 2. To determine the complex-formation constants of the species that are formed at the most acidic pH values, it is necessary to use a competition method, which exploits the presence of a second ligand (or another metal) whose speciation pattern is fully known under the same experimental conditions. Savastano et al. [12], after having initially explored (and then discarded) the possibility of using EDTA as a competitor, decided to revisit the hydrolysis equilibria of zirconium, in order to exploit the competition between DFOB and the hydroxyl ion. The complex-formation equilibria between Zr(IV) and DFOB were thus studied by potentiometry in the pH range 2.5–11.5, and were confirmed by spectroscopic investigations using small-angle X-ray scattering (SAXS) and MALDI mass spectrometry. The formation of very stable mononuclear complexes was confirmed but, as an unexpected result, the formation of dinuclear species (see Figure 3a), predominant at neutral pH even in very dilute solutions, was also detected. The coordination geometry was investigated through DFT calculations: regarding the 2:2 complexes, the lower energy conformer was described as a “barbell”, in which two metal centers surrounded by the donor atoms form the spherical terminals.

The potentiometric technique and the competition with the hydroxyl ion were also used by Toporivska et al. [13] to study the complex-formation equilibria between Zr(IV) and DFOB between pH 2.3 and 11.0, in the presence of an excess of ligand. However, in this case, the speciation model, better explaining the potentiometric results, contained only variously protonated mononuclear complexes (see Table 6 and Figure 3b). ESI-mass spectra confirmed the presence of only mononuclear species. According to the DFT calculations of Holland et al. cited above [30], it has been hypothesized that all hydroxamic groups are already deprotonated and bound to zirconium at pH 2.3 in the complex [MHL]^+^, and that the terminal amino group is free and protonated. When pH is increased, the first deprotonation is probably due to a water molecule linked to zirconium to complete the coordination sphere, while the second deprotonation has been attributed to the terminal amino group without any interaction with the metal, as in the case of Fe(III) and Ga(III) (see above). To more deeply investigate the behavior of the system at more acidic pH, spectrophotometric measurements of competition with the Fe(III) ion were also performed. In fact, the speciation model of the Fe(III)/DFOB system is known (see above), and the Fe(III) complexes are intensely colored (in contrast to Zr(IV)). Solutions of Zr(IV)/DFOB were titrated with Fe(III), those of Fe(III)/DFOB were titrated with Zr(IV), and those of Fe(III)/Zr(IV) were titrated with DFOB, finding results in excellent agreement with each other. The average value of the constant of the [MHL]^2+^ complex (47.6) was somewhat higher than, but in reasonable agreement with, that obtained by potentiometry (46.4). Differential isothermal calorimetry (ITC) measurements also confirmed a 1:1 stoichiometry at very acidic pH.

The complexity of the system under examination, both from the thermodynamic and kinetic point of view, partly justifies these differences. However, they can become important for the preparation of radiopharmaceuticals and further investigations would be welcome in order to definitely clarify the complex-formation behavior between Zr(IV) and DFOB.

### 2.6. DFOB Complexes with High Valence Metal Ions

The ability of polyhydroxamate ligands to stabilize highly charged metal ions in solution inspired several research groups to study the complex-formation equilibria of DFOB with high valence metals, such as those shown in Table 7. In the case of vanadium, it has been shown that, at the most acidic pH values, there is a reversible displacement of the oxygen bound to the metal to form “nonoxo” complexes in which the “naked” metal ion is directly coordinated to the three hydroxamic groups of DFOB [8]. As the pH increases, oxygen atoms replace one or two hydroxamic functions to form the classical complexes of V(IV)O^2+^ or V(V)O_2_^+^.

A combined potentiometric, spectrophotometric and NMR method was employed to study the complex-formation equilibria of DFOB with Th(IV) [33]. The investigation demonstrated the formation of highly stable and soluble mononuclear complexes, capable of mobilizing thorium even in the presence of other competing metal ions such as Fe(III). DFOB has been suggested as an additive for liquid/liquid or solid/liquid extractions with application in environmental remediation. Finally, the formation of complexes between Pu(IV) and DFOB in solution has been studied by Jarvis and coworkers [34] through spectrophotometry (see Table 7).

**Table 7 molecules-26-03255-t007:** Overall complex-formation constants (log*β*) of DFOB with high valence metal ions, under different experimental conditions.

		[MH_3_L]^4+^	[MH_2_L]^3+^	[MHL]^2+^	[ML]^+^	*I* (mol·dm^−3^)	Background Electrolyte	*T* (°C)	Ref.
V(IV)		40.13	37.09	29.66	-	0.60	NaCl	25.0	[8]
Th(IV)		-	36.40	34.40	26.60	0.10	KCl	25.0	[33]
Pu(IV)		-	-	-	30.80	0.10	KCl	25.0	[34]
	[MH_5_L]^7+^	[MH_3_L]^5+^	[MH_2_L]^4+^	[MHL]^3+^					
V(V)	45.0	42.46	37.54	28.74		0.60	NaCl	25.0	[8]

### 2.7. Overall Comparison among Metal-Binding Ability towards DFOB

Figure 4 shows the correlation between the formation constant of the 1:1 non-protonated DFOB complexes with all of the above metal ions and the corresponding formation constants of the hydroxyl complex MOH^n+^. As already noted by Evers et al. [7], these constants are well correlated with each other, suggesting the same coordination mode for all the metal/DFOB complexes. This graph allows us to estimate with a good approximation the DFOB formation constant with any metal, simply starting from the knowledge of its hydrolysis equilibria. In other words, as it generally holds true for ligands containing only negatively charged oxygen donor-atoms, the selectivity towards different metals is regulated by the corresponding hydrolysis behavior. To modulate the selectivity of this type of ligands, it is possible to modify their structure, e.g., making it more rigid or introducing other donor atoms (or replacing negatively charged oxygen donors), such as nitrogens or even neutral oxygens [7].

A further method of comparing the stability of the different DFOB metal complexes is based on the two parameters pM and *K*_D_. pM is defined as the negative logarithm of the free metal ion concentration in a solution containing a total amount of 10^−5^ moles·dm^−3^ of the ligand and 10^−6^ moles·dm^−3^ of the metal ion, at pH 7.4 [37]. *K*_D_ is the dissociation constant referring to the hypothetical equilibrium: ML = M + L, where M represents the metal ion, L is the ligand and the charges are omitted for simplicity. The dissociation constant can be written as *K*_D_ = [M]·[L]/[ML], where the square brackets signify the molar concentration of the reactants. *K*_D_ can be considered a conditional constant, and its dimensions are considered to be those of a molarity; the physical meaning of *K*_D_ is the molar concentration of free metal when the ligand is 50% in its free form in solution and 50% bound to the metal, regardless of the stoichiometry of the formed species [38]. The *K*_D_ value can be experimentally obtained by titrating a solution of known ligand concentration with a standard solution of the metal ion and monitoring the residual concentration of the free ligand or that of complexed ligand. In the normally used ligand-concentration range (1–0.001 mmol·dm^−3^), the *K*_D_ value does not depend on the total concentration of ligand, except in the case of the formation of polynuclear oligomers in solution. Both pM and *K*_D_ can be calculated from the speciation models reported in Table 1, Table 2, Table 3, Table 4, Table 5, Table 6 and Table 7; in the case of *K*_D_, the titration of a DFOB solution with the considered metal ion can be simulated [38].

The calculated values of pM and *K*_D_, shown in Table 8, are well correlated with each other, except for two specific situations: (*i*) the speciation model of DFOB complexes with Zr(IV) suggested by Savastano et al. [12] contains oligomeric species and the *K*_D_ value considerably decreases as the concentration of the ligand in solution increases, thus not providing an univocal value; (*ii*) in the case of the weakest complexes, pM is fixed at the value of 6, which means that, at pH 7.4, all the metal is free and no complex forms. This is the case of the alkaline earth metals. In fact, since the pM value is calculated for a total metal concentration of 10^−6^ mol·dm^−3^, the pM value cannot fall below 6. The values of pM and *K*_D_ of Table 8 are also in good agreement with the graph of Figure 4, even if the metal sequence is not perfectly identical. The charge of the metal ion certainly has an important effect on the stability of the DFOB complexes. The most stable complexes are those formed by Zr(IV), and this justifies its use as a chelator of ^89^Zr employed in diagnostic imaging. The high stability of Fe(III) complexes with DFOB is well known and is the basis of its biological activity as a siderophore and of its use in chelation therapy. The plutonium complexes are also very stable, and this makes DFOB a potential efficient antidote in cases of intoxication with this very toxic metal.

## 3. DFOB in Medicinal Chemistry and Other Applications

During the past few years, DFOB has been widely investigated for its potential use in many applications, from the biological to the environmental field. It is an extremely versatile molecule with an impressive number of applications. The “fortune” of DFOB must be firstly sought in its appreciable biocompatibility and modest cost of production. Although small-scale chemical synthesis in solution and solid-phase is possible, nowadays DFOB industrial-scale production occurs through the efficient bacterial fermentation of an overproducing strain of *S. pilosus*. Moreover, the solubility in water and other organic solvents makes it suitable for derivatization reactions mostly occurring on its terminal amine group, which is not involved in metal complexation and can be successfully employed to graft small molecules, antibiotics and other functionalities, in order to confer new properties to the DFOB molecule and broaden its scope [39].

Despite the clear advantages for the use of DFOB in numerous human diseases, including, above all, iron overload-associated pathologies, the administration of this siderophore may give rise to some toxic effects for the patients’ health, such as cardiovascular, respiratory, gastrointestinal, renal, cutaneous and nervous systems diseases, growth failure and bone dysplasia, hearing impairment, retinal pigment epithelium changes, visual loss and impaired night vision. Furthermore, the main limit of DFOB as a therapeutic agent comprises its parenteral administration and associated problems with adherence [40,41], and therefore huge efforts are being devoted to the design of DFOB derivatives with improved therapeutic potentialities. Without any doubt, the main application of DFOB concerns the treatment of diseases associated with iron homeostasis imbalance, and in particular iron overload; nevertheless, the importance of iron in many biological processes and its involvement in oxidative stress make DFOB an outstanding molecule that may be studied and employed for different therapeutic and diagnostic purposes.

### 3.1. DFOB-Mediated Iron Chelation Therapy

DFOB and, in general, chelating drugs can be used as main, alternative or adjuvant therapy to counteract several human diseases. The chelation therapy primarily finds application in metal detoxification, but can also display promising antioxidant, anticancer and antimicrobial effects [42,43]. For about three decades, DFOB was the only approved drug to treat secondary iron overload disease, and it is currently widely used to reduce iron accumulation and deposition in tissues, which cause many diseases and organ dysfunctions, including arrhythmias, congestive heart failure, diabetes, endocrinopathies and hepatic fibrosis. The mesylate salt of DFOB, known as Desferal^®^, is currently considered an “Essential Medicine” according to the World Health Organization (WHO). The mechanism of action of the DFOB drug is to primarily bind Fe(III) in the vascular space and to form the corresponding complex FOB, which is subsequently excreted in urine or bile [44]. The most important factor of free iron overload toxicity is related to its participation in Fenton-type redox reactions and to the production of free radicals and reactive oxygen species (ROS), such as the highly reactive hydroxyl radical (OH∙) or lipid radicals. Moreover, free iron ions have the ability to access highly vascular tissues, especially hepatic, cardiac and endocrine cells, being extremely harmful for the corresponding organ function [45].

The importance of DFOB as an iron-chelating agent to treat secondary hemochromatosis is undoubtedly well known and confirmed by several studies and clinical evidence. The DFOB-mediated iron chelation therapy (DFOB-ICT) commonly applies to both transfusion-dependent anemias, such as β-thalassemia and sickle cell anemia, and non-transfusion-dependent thalassemia or myelodysplastic syndromes [46]. DFOB is also used in combination with other chelating agents or adjuvants, in order to increase the efficacy of iron excretion and reduce adverse effects arising from the DFOB monotherapy. Certainly, ascorbic acid plays an important role in DFOB-ICT and it is considered as a standard adjuvant therapy: several studies highlight that the administration of oral ascorbate in combination with subcutaneous DFOB causes a substantial increase in iron excretion (up to 245% in thalassemia patients) [47,48,49]. Another well-known example of combination therapy for iron chelation is that with DFOB and deferiprone, a newer, orally administrable, iron chelating drug. The additional administration of deferiprone is particularly effective against cardiac iron overload, thanks to the more efficient access to cardiac myocytes of this synthetic agent, which subsequently transfers the bound iron to the DFOB located in plasma [50,51]. Further pilot clinical trials also revealed promising results for the DFOB/deferasirox combination therapy [52].

### 3.2. DFOB Antioxidant Activity

Although the most important property of DFOB is its ability to reduce the concentration of excess iron in the organism, an additional effect is related to the inhibition of iron catalyzed production of oxygen radical species. DFOB-ICT can therefore find application in several diseases where oxidative stress contributes to the pathophysiology; examples include renal and hepatic diseases [45], neurodegenerative diseases [53], and even the prevention of oxidative stress-induced apoptosis in stem cells [54]. Moreover, since the phenomenon of oxidative stress greatly contributes to lead (Pb)-mediated toxicity, a recent study reveals promising preliminary results on the antioxidant activity of DFOB to protect against Pb-induced cardiotoxicity in rats [55].

Particular attention is given to nervous system disorders (neurodegenerative disorders, such as Alzheimer’s disease and Parkinson’s disease, stroke, brain injury), where ferroptosis—the iron-dependent programmed cell death characterized by accumulation of free iron and ROS species, with the consequent peroxidation of polyunsaturated fatty acids in the plasma membrane—represents a major implication [56]. DFOB displays a promising therapeutic antioxidant activity, being able to cross the blood–brain barrier and to reduce the iron content in various brain regions [40]. Moreover, in animal models, it has been demonstrated to induce neuroprotection and to reduce hemorrhagic brain injury [53,57,58]. The research group of Xi [59] demonstrated that DFOB can attenuate perihematomal iron accumulation, white matter injury and neuronal death, but it also slows hematoma resolution in animal models of intracerebral hemorrhage [58,59]. The latter effect has been also confirmed by Cao et al., since DFOB proved to reduce the formation of terminal complement complex and to attenuate erythrophagocytosis [60]. Nevertheless, some further clinical evidence suggests the beneficial role of DFOB in the treatment of intracerebral hemorrhage patients, but further investigations are required to claim its effective therapeutic use in humans [61].

### 3.3. DFOB Anticancer Activity

DFOB also exhibits anticancer activity withdrawing iron ions from cancer cells [62]. Iron, in fact, contributes to cancer cell growth and proliferation, and cellular accumulation of this metal ion has been shown to favor the onset of the most aggressive forms of cancer [63]. As a consequence, iron chelators have been studied as cancer chemopreventive and chemotherapeutic agents. DFOB, thanks to its clinical safety and documented efficacy in chelation therapy, was the first explored molecule in this context and its administration as supportive anticancer therapy has been tested. It appeared to interfere with DNA synthesis and to inhibit the progression of skin cancer, breast adenocarcinoma, hepatocellular carcinoma and ovarian cancer [63,64,65,66,67]. However, the promising anticancer activity of DFOB is still limited and further studies are required to elucidate its potential benefit as adjuvant therapy in the treatment of malignant tumors. Moreover, the use of DFOB as anticancer agent presents some drawbacks related to its low membrane permeability and short plasma half-life. This has led to focus on different iron chelators, characterized by higher lipophilicity, membrane permeability and selective activity [68].

### 3.4. DFOB Potential Anti-SARS-CoV-2 Activity

The potential application of iron chelation through the administration of DFOB in order to fight SARS-CoV-2 infection was recently hypothesized by Dalamaga et al. [69]. Several studies highlight the critical role of iron in the replication of RNA viruses such as HCV, HIV and West Nile virus [70,71,72,73]. Pulmonary iron accumulation is also known to favor the progression of respiratory diseases like pulmonary fibrosis and acute respiratory distress syndrome [69]. Therefore, DFOB may show antiviral properties by reducing the bioavailability of iron ions. In addition, DFOB appears to have immunomodulatory effects in vitro and in vivo [72]. Some reports also showed high serum ferritin levels in COVID-19 patients, and an increased amount in deceased patients with respect to COVID-19 survivors [73,74]. High concentrations of serum ferritin are often associated with cellular iron imbalance and to oxidative stress and inflammatory response, suggesting that iron chelation therapy may benefit COVID-19 patients.

### 3.5. Other Therapeutic Applications of DFOB

The importance of iron for the survival of all living beings and its involvement in many biological processes confers to DFOB and other iron chelators the potential to expand their therapeutic properties to several pathologies. The iron chelation therapy may help in treating or preventing atherosclerosis, by hampering the formation of atherosclerosis lesions [75,76,77]. Iron depletion induced by DFOB has been also investigated as an antimalarial strategy [78]. The potential use of DFOB in the treatment of chronic inflammation, such as in diabetes, is also well documented. Some studies reveal an association between the occurrence of iron overload diseases and type-2 diabetes, together with an increased risk of diabetic complications [79,80]. The benefit of iron chelation therapy is likely associated with the improvement of coronary microvasculature function [81]. A work by Dongiovanni et al. [82] showed that DFOB not only induced iron depletion in HepG2 hepatocytes and in rat liver, but it also increased glucose metabolism and upregulated insulin receptor activity and signaling. The potential use of DFOB for the treatment of Type-1 diabetes has been also hypothesized by Najafi et al. [83] on the basis of pre-clinical tests where DFOB appeared to enhance the mobilization and homing of mesenchymal stem cell, increasing the hypoxia-inducible factor 1α (HIF-1α) activation. Other studies report the effectiveness of DFOB as HIF-1α inducer and stabilizer through iron depletion, thereby promoting neovascularization and enhancing angiogenesis and wound maturation in mice. The simultaneous effects as iron chelator also include the prevention of oxidative stress and make DFOB a promising therapeutic agent to improve the healing of chronic wounds, with a pro-angiogenesis function [84].

### 3.6. Radiometal and Optical Imaging

DFOB can be used to develop diagnostic methods for detecting infections and/or for theranostic applications. Its ability to stably bind different metal ions represents an outstanding property to produce radioactive complexes that can be theoretically absorbed by the target cells, generating a radioactive substrate accumulation. In the case of pathogenic (namely bacterial or fungal) cells, the response could even vary according to the severity of the infection, providing a high selective and effective diagnostic tool. There are two usually applied methods to design a DFOB-based imaging agent, namely: *(i)* substituting Fe(III) with a radioactive ion or *(ii)* derivatizing the DFOB molecule coupling an additional functionality to the siderophore scaffold.

Since no isotopes of iron with suitable properties for nuclear imaging in terms of half-life and photon emission exist, the first investigations of radiolabeled DFOB metal complexes were focused on ^67^Ga(III) and ^111^In(III) [85,86]; both of these isotopes are gamma-emitting radionuclides and are widely used in nuclear medicine for single photon emission computed tomography (SPECT) imaging. A recent study of Petrik et al. also shows the promising application of ^68^Ga(III)/DFOB complex for molecular imaging by the PET technique in animal models. These results also revealed the potential application of the ^68^Ga(III)/DFOB complex for the molecular imaging of different bacterial infections, including *Pseudomonas aeruginosa* and *Staphylococcus aureus* [87].

DFOB zirconium labelling has also been studied as a promising clinical tool for immuno-positron emission tomography (immunoPET), since the predisposition of DFOB to be derivatized at its terminal amino moiety allows the conjugation of a variety of targeting vectors (e.g., monoclonal antibodies or antibody fragments) [88]. Zr(IV) complexation by DFOB occurs in a 1:1 ratio, with less than 2% demetallation after 7 days in serum; moreover, DFOB conjugates exhibit low immunogenicity. However, despite these attractive properties, preclinical trials with ^89^Zr(IV)/DFOB-conjugates reported accumulation of the osteophilic radiometal in bones from 3% to 15% after one week. The nature of the radioactive species accumulating in the bones remains uncertain as well as the clinical implications of ^89^Zr persistence in the organism. The insufficient stability of ^89^Zr(IV)/DFOB complex is attributed to DFOB’s inability to fully saturate the coordination sphere of the octavalent metal ion. In order to overcome such limitations, a family of novel DFOB derivatives with higher metal binding affinity for Zr(IV) was developed. Patra et al. synthetized an octadentate tetrahydroxamate DFOB analogue termed DFO*, which exhibits a higher stability profile than the natural hexadentate siderophore, but limited water solubility. This latter problem led to the development of an oxygen-containing analogue, termed oxoDFO*. Recently, both DFO* and oxo-DFO* were studied in terms of ^68^Ga(III)-radiolabeling. Although they represent potential alternative chelators for ^68^Ga(III), quantitative radiolabeling required elevated temperatures and the formed complexes were significantly less stable than the corresponding ^89^Zr(IV)-complexes [89]. More recently, a number of promising new Zr(IV) DFOB-based chelators with higher radiolabeling performance and in vivo stability were reported; they include DFOcyclo*, which contains a fourth cyclic hydroxamic acid group [90], and DFO2, which is obtained by tethering two DFOB molecules together [91]. Certainly, ^89^Zr(IV)-conjugates have been gaining significant importance in nuclear medicine for the diagnosis of different types of cancer and this makes DFOB and its analogues important clinical immunoPET imaging probes; some representative clinical-stage ^89^Zr(IV)/DFOB-conjugates include: ^89^Zr(IV)/DFOB-J591, targeting metastatic prostate cancer, ^89^Zr(IV)/DFOB-trastuzumab and ^89^Zr(IV)/DFOB-pertuzumab, for imaging HER2/neu expression, and ^89^Zr(IV)/DFOB-bevacizumab, for imaging vascular endothelial growth factor [29,39,88,92].

As previously mentioned, DFOB derivatives have been widely investigated to improve the potentiality of the original molecule. The coupling with bioactive molecules—not only antibodies but also nanoparticles, albumin, fibrinogen and other proteins—allows the formation of bifunctional agents able to exhibit radio-imaging properties and, according to the added functionality, therapeutic or targeting activity [86].

Aside from the radio-imaging application for SPECT, PET and immunoPET, several efforts have been made in order to design novel DFOB-based tool for the detection of infections through optical imaging. Taking advantage of the siderophore-mediated iron transport mechanism, it is possible to selectively deliver an optical probe inside the target cell and monitor the iron acquisition mechanisms in different bacteria and fungi. To achieve this purpose, DFOB is therefore covalently bound to a fluorescent molecule through its amino group [86,93]. An example is given by the fluorescent derivative 7-nitrobenz-2-oxa-1,3-diazole-deferoxamine B (NBD-DFOB) [94], which has been employed to detect the iron acquisition pathway in pathogenic fungi of the order Mucorales. Other fluorescent DFOB derivatives include 2-cyanonaphtho [2,3-c]-2*H*-pyrrolyl-DFOB and various fluorescein-DFOB-conjugates [1].

### 3.7. Sensing Devices

The ability of DFOB to coordinate Fe(III) and other metal ions makes it suitable for metal detection and sensing, with application in both biological and environmental fields. In principle, the analytical information can be obtained by optical or electrochemical signals. Many sensors have been developed, and the rationale behind them is once again the derivatization of the DFOB molecule with different functionalities; some examples include the coupling with fluorophores, nanoparticles and nanotubes, mesoporous silica films and optical fibers. Electrochemical devices, meanwhile, generally require the covalent mounting of the DFOB on the electrode surface (Table 9). DFOB, with respect to other kind of siderophores, has been mainly employed for the detection of Fe(III), and only one study from Karimi Shervedani et al. [95] highlighted its analytical application as voltammetry sensor for Ga(III). Some innovative approaches are also under investigation for the detection of Fe(III) and V(V) ions by colorimetric assays of paper-based sensors [96].

DFOB can be also employed for pathogen detection and sterilization techniques by means of bacteria capture platform technology. The detection mechanism is based on the siderophore biological function of iron shuttle, which can be exploited to recognize and bind specific bacterial iron receptors (i.e., FepA and FhuA). Kim et al. developed a selective siderophore-inspired bacteria capture platform, where the DFOB secreted by *Yersinia enterocolitica* is conjugated with bovine serum albumin and immobilized on a gold electrode surface. The analytical information is obtained by dark-field microscopy analysis of the scattered light, followed by Fourier transform analysis of the scattering pattern [106]. This quite recent and innovative strategy for bacteria detection opens the way to novel analytical applications for DFOB and other siderophores, which can be employed as highly selective ligands for rapid detection and identification of a cognate pathogen [39].

### 3.8. Trojan Horse Strategy

The importance of DFOB as microbial iron transporter not only has interesting implications for potential bacterial sensing or for the diagnosis of infections but, and above all, it enables designing novel potential therapeutic agents against pathogenic microorganisms. The so called “Trojan Horse” strategy, which is inspired by the natural antibiotics sideromycins, can be exploited to circumvent the cell membrane-associated drug resistance and to selectively deliver an antimicrobial agent inside the pathogen cells, taking advantage of the siderophore-mediated iron transport across the membrane. The conjugated system structurally consists of the iron-chelating component which is covalently bound to the pharmacophore molecule through a linker moiety, so that the iron-siderophore complex is internalized inside the cell together with the conjugated pharmacophore. This approach has been widely investigated and a number of DFOB conjugated systems have been proposed and tested against many pathogens, including both bacteria and fungi [1,39,93]. Conjugation with another chelating agent or metal-complex is also possible, as recently reported for DFOB conjugates with ruthenium(II)-arene fragments, which display potential dual antibacterial/anticancer application [107].

A slightly different Trojan horse antimicrobial approach consists, instead, of the substitution of iron with another (preferentially toxic) metal ion in the siderophore complex, thus avoiding the synthesis of conjugated systems. Banin et al. investigated this approach in 2008 and tested the antimicrobial activity of the non-radioactive Ga(III)/DFOB complex against *P. aeruginosa*, showing that the complex is able to bind iron, reducing its supply for the pathogen cells, and that it simultaneously releases Ga(III) ions, which compete with the isosteric Fe(III) for the binding to several proteins, thus interfering with iron metabolism [108]. This antimicrobial effect showed promising results and may be of interest for the design of other antimicrobial agents aimed at disrupting metal ion homeostasis. Similar studies were later performed with other DFOB metal complexes. Mattos et al. reported that Cd(II)/DFOB complex is a more efficient antimicrobial agent than the CdCl_2_ salt [109]. Other studies on Al(III) and Ga(III) revealed that their complexes with DFOB (and DFOB-caffeine derivative) enhance or at least maintain the antimicrobial activity of free metal ions and free siderophore against *Escherichia coli*, *P. aeruginosa*, *S. aureus* and *Candida albicans*; nonetheless, their biological activity requires further improvement for competitive application in the treatment of infections [110].

## 4. Conclusions

This review collects and summarizes the available data on the complex-formation equilibria in solutions of DFOB, a bacterial siderophore, with different metal ions. The reported thermodynamic data highlight the potential use of DFOB as an effective metal chelator, not only for the Fe(III) ion, but also for many other cations, including several transition, lanthanoid and actinoid metal ions. A comparison in terms of stability among more than 20 metal complexes with DFOB has been reported. According to the calculated pM and *K*_D_ values for DFOB metal complexes (see Table 8), Zr(IV) ion is able to form the most stable species, trivalent metal ions generally exhibit a higher tendency to stabilize DFOB complexes with respect to divalent ones, and alkaline earth cations form eventually the less effective complexes. The entirety of the collected thermodynamic results on the DFOB complex-formation equilibria reflect, therefore, the extreme versatility of DFOB applications. The efficacy of DFOB as metal chelator makes it suitable for many innovative medicinal uses, moving beyond the iron chelation therapy. Many efforts are now being devoted to the use of DFOB in radiometal imaging, including PET, immunoPET and SPECT techniques, thanks to the ability to form stable complexes with Zr(IV), Ga(III) and In(III) radioisotopes. The ability to bind very toxic heavy metal ions, for instance Pu(IV) and other lanthanoid or actinoid ions, also makes DFOB a potential efficient antidote in case of severe intoxication. For the same reason, this molecule can find application in environment remediation or as metal sensing device. Another important element of the DFOB molecule is its terminal amino group, not participating in metal complexation. The amine moiety can be relatively easily derivatized with small molecules, chromophores, pharmacophores or even antibodies and other functionalities, in order to improve the DFOB properties and expand its fields of application. Despite its “old age”, without any doubt, DFOB is nowadays a widely explored drug, which still has a lot to offer.

## Data Availability

The data presented in this study are available on request from the corresponding author.

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
