# Peer review of "Deferoxamine B: A Natural, Excellent and Versatile Metal Chelator"

_molecules, 2021, doi:10.3390/molecules26113255_

Round 1
Reviewer 1 Report
In the manuscript, the authors collecting and summarizing available data of the complex-formation equilibria of Deferoxamine B (DFOB) with different metal ions in solution and discussing application of DFOB in medicinal chemistry in the past decade. Basically, the logic of the manuscript is smooth However, there are several things that need to be clarified and possibly modified in order to make the manuscript easier to read.
- In the “Introduction” part, the authors describe the structure and functional group composition of DFOB. Here, the functional groups in the “Scheme 1” can be expressed and labeled, which will make it much clearer for the reader if corresponding to the words. In addition, it is mentioned that DFOB and Fe (III) form very stable octahedral complex FOB. It is highly recommended to illustrate it by scheme.
Similarly, it would be better if the descriptions of structures such as the free ligand, completely deprotonated form L3-, fully protonated form H4L+, and others, in “Section 2.1” are accompanied by scheme(s).
- The constant data in parentheses in the Table is a little hard to understand. For example, Table 2 states that “The constant in parentheses corresponds to the formation of the 1:1 complex of Fe (III) with DFOB protonated at its amino group and therefore refers to the equilibrium: Fe3+ + HL2- = MHL+.” But this value is labeled under [ML], so it is not clear what the value in parentheses mean.
- In the part “2.3 DFOB complexes with divalent metal ions”, it is said that “The most stable complexes are those with Cu (II), which is also the only metal in the series capable of forming a binuclear species.” and “Complexes of Sn (II) with DFOB are also more stable than those of Cu (II) and the formation of two binuclear species has been proposed.” It is not clear which one of these two metal complexes is more stable and what data this conclusion is based on. Also, there is no reference why that Sn (II) complex has two types of binuclear species.
There are many similar questions throughout the manuscript, which may require the authors to provide explanations in detail about what data/numbers the concluding statements are derived from.
- In part “2.5 DFOB complexes with Zr (IV)”, it is stated that “The great tendency to hydrolysis of zirconium in aqueous solution is a significant obstacle to the thermodynamic study of these systems and only very recently the problem has been independently addressed by two research groups with results that are not entirely in agreement with each other, as shown in Table 6.” But it is hard to see why their studies are not in agreement with each other in the Table 6.
- Label error, “3.3” might be changed to “3. DFOB in medicinal chemistry and other applications”. Also, it is recommended to add the Conclusion section to the manuscript in order to make it more completed and provide some outlines for observed trends.
Reviewer 2 Report
This was an interesting review; however, I am uncertain that it presents any data in a new way or adds any new interpretations to the field.
Numerous minor grammatical errors distract from the text. Most notable are compound sentences lacking the required commas.
Several figures have literature citations but lack any indications of receiving permission from the copyright holders to reproduce.
The review must have a figure of the structure of FOB or a similar figure with Mn+ instead of Fe3+.
The caption to Figure 13 fails to indicate that part B was done in the presence of excess ligand as described in the text. Frankly, I think the authors might be more critical of the study in Ref. 12.
